# Phosphoinositide 3-Kinase (PI3K) Inhibitors and Breast Cancer: An Overview of Current Achievements

**DOI:** 10.3390/cancers15051416

**Published:** 2023-02-23

**Authors:** Alexandre Bertucci, François Bertucci, Anthony Gonçalves

**Affiliations:** Medical Oncology Department, CRCM, INSERM, CNRS, Institut Paoli-Calmettes, Aix-Marseille University, 13009 Marseille, France

**Keywords:** breast cancer, PI3KCA, pan-PI3K inhibitor, selective PI3K inhibitor, resistance mechanism, overview, target therapy

## Abstract

**Simple Summary:**

Breast cancer remains the fourth-leading cause of death worldwide, and therapeutic improvement is warranted. The phosphatidylinositol 3-kinase (PI3K) pathway is one of the major pathways in oncogenesis, and PI3K alterations are common in all breast cancer subtypes. Despite modest clinical activity and a high toxicity rate, pan-PI3K inhibitors paved the way for selective PI3K inhibitor development. In this overview, we cover the past, the present, and potential paths, as well as the therapeutic challenges to come for this therapeutic class.

**Abstract:**

The phosphatidylinositol 3-kinase (PI3K) pathway is one of the most altered pathways in human cancers, and it plays a central role in cellular growth, survival, metabolism, and cellular mobility, making it a particularly interesting therapeutic target. Recently, pan-inhibitors and then selective p110α subunit inhibitors of PI3K were developed. Breast cancer is the most frequent cancer in women and, despite therapeutic progress in recent years, advanced breast cancers remain incurable and early breast cancers are at risk of relapse. Breast cancer is divided in three molecular subtypes, each with its own molecular biology. However, PI3K mutations are found in all breast cancer subtypes in three main “hotspots”. In this review, we report the results of the most recent and main ongoing studies evaluating pan-PI3K inhibitors and selective PI3K inhibitors in each breast cancer subtype. In addition, we discuss the future of their development, the various potential mechanisms of resistance to these inhibitors and the ways to circumvent them.

## 1. Introduction

Worldwide, in 2020, breast cancer (BC) represented the most frequent cancer, with approximately 2.3 million cases, and the fourth-leading cause of death, with approximately 685,000 deaths [1]. In early BC, which represents approximately 90% of cases, the treatment is based on surgery, radiotherapy, and, depending on the molecular subtype, on endocrine therapy (ET), chemotherapy, targeted therapy, and immune therapy. In advanced BC, the standard of care is based on systemic therapy and varies according to the molecular subtype. When the human epidermal growth factor (HER2) is overexpressed (HER2+), the treatment includes a combination of either cytotoxics (or less commonly ET) with anti-HER2 agents, including anti-HER2 monoclonal antibodies trastuzumab and pertuzumab, or tyrosine kinase inhibitors, such as lapatinib or tucatinib. In trastuzumab-resistant disease, antibody–drug conjugates (ADCs), such as trastuzumab emtansine and, most recently, trastuzumab deruxtecan, are recommended [2]. In triple-negative breast cancers (TNBCs), the standard of care is chemotherapy, which may be combined with an immune checkpoint inhibitor when PD-L1 is expressed, and, more recently, ADCs, such as sacituzumab govitecan or trastuzumab deruxtecan. In the Estrogen Receptor (ER)/Progesterone Receptor (PR)-positive/HER2-negative subtype (hormone receptor HR+/HER2−), the treatment relies upon ET in combination with a CDK4/6 inhibitor [3,4,5]. The therapeutic management is characterized by multiple lines of ET, possibly in combination with other targeted therapies, until the appearance of a clear ET resistance, followed by the use of palliative chemotherapy, which may include trastuzumab deruxtecan in the case of HER2 low expression. In both advanced stages of TN and HR+/HER2− subtypes, PARP inhibitors (olaparib, talazoparib) are also an option in the case of germline *BRCA1* or *BRCA2* mutations [6,7,8,9].

While very significant progress, including an improvement in overall survival, has been made during the last two decades in all the molecular subtypes, advanced BC is still considered as an incurable disease and the disease recurrence almost invariably occurs after periods of remission on specific systemic treatments. In this context, several attempts have been made to improve the overall efficacy of the already established standard of care. Amongst them, the therapeutic modulation of the phosphatidylinositol 3-kinase (PI3K)/Akt/mammalian target of rapamycin (mTOR) pathway has been one of the most investigated strategies to improve the survival at advanced stages of the disease in each molecular subtype. After various generations of mTOR inhibitors failed to demonstrate an improvement in overall survival, and while Akt inhibitors are currently under comparative evaluation, PI3K inhibitors have been developed during the last 10 years, mitigating clinical achievements. In this review, we examine the main features of this important biological pathway and the rationale for its therapeutic targeting in BC, as well as the main clinical results that have been obtained and the next perspectives for further clinical experimentations.

## 2. The Phosphatidylinositol 3-Kinase (PI3K) Pathway Molecular Alterations in Breast Cancer and the Rationale for Therapeutic Targeting

The PI3K pathway is one of the most frequently altered pathways in human tumors [10]. PI3K belongs to the family of intracellular lipid kinases and can be divided into three classes. Class I is activated by insulin or other growth factors and can be further divided into classes IA and IB. Class IA PI3K is associated with tyrosine kinase receptors (RTKs) and class IB is associated with heterotrimeric G-protein-coupled receptors (GPCRs). Both are composed of a catalytic subunit and a regulatory subunit. For class IA PI3K, which is the most directly involved in carcinogenesis and is encoded by the *PIK3CA* gene, the catalytic subunit isoforms are p110α, p110β, p110γ, or p110δ. The catalytic subunit associates with the regulatory subunit (encoded by *PIK3R1*, *PIK3R2,* or *PIK3R3*) [11,12], which inhibits the kinase activity of p110, stabilizes p110, and promotes a negative retrocontrol [11]. When an RTK is activated by growth factors, PI3K is recruited to the plasma membrane and is activated, thus phosphorylating phosphatidylinositol 4,5 biphosphate (PIP2) to generate phosphatidylinositol 3,4,5 triphosphate (PIP3). PIP3 recruits and binds AKT or other proteins with a pleckstrin-homology (PH) domain such as 3-phosphoinositide–dependent protein kinase 1 (PDK1). Then, AKT is phosphorylated and activated by PDK1 and mammalian target of rapamycin complex 2 (mTORC2) [13], triggering several phosphorylation-based signaling cascades. PI3K plays a central role in several cellular processes, including DNA synthesis, metabolism, and action on actin cytoskeleton [14,15]. The tumor suppressor phosphatase and tensin homolog (PTEN) negatively regulates the action of PIP3 via 3′-phosphatase activity, thus transforming PIP3 into PIP2 [16], which stops the phosphorylation cascade (Figure 1).

Class IA features tyrosine kinase receptors (RTKs) and class IB has G-protein-coupled receptors (GPCRs). When a growth factor (L) or a chemokine (C) activates the receptor, PI3K is recruited to the cellular membrane and is activated, thus phosphorylating phosphatidylinositol 4,5 biphosphate (PIP2) to generate phosphatidylinositol 3,4,5 triphosphate (PIP3). PIP3 recruits and binds to protein kinase B (AKT). Then, AKT is phosphorylated and activated by 3-phosphoinositide-dependent protein kinase 1 (PDK1) and mammalian target of rapamycin complex (mTOR), triggering several phosphorylation-based signaling cascades. PI3K plays a central role in several cellular processes, including DNA synthesis, cellular growth, metabolism, survival, and cell motility. 

Alteration of the PI3K pathway is one of the most common genetic alterations in human cancers [17]. Each actor in the pathway may be concerned with different alterations (mutations, deletions, or amplifications), with *PIK3CA* and *PTEN* at the top of the list and being, respectively, the second and third most highly mutated genes in human cancers [10,18,19]. The *PTEN* tumor suppressor gene [20,21] is commonly inactivated in sporadic cancers. PTEN loss induces an increase in PIP3, resulting in a constitutive activation of the PI3K pathway, which promotes cell proliferation and cell cycle progression. *AKT1* gain-of-function mutations may also be found and similarly activate the pathway. *PIK3CA* mutations are found in approximately 30% of different solid tumors [19,22,23]. The catalogue of somatic mutations in cancer (COSMIC) shows that 80% of *PIK3CA* mutations concern only three “hotspots” and are located on the kinase and helical domains of the catalytic subunits. These “hotspots” H1047R, E542K, and E545K may be found in a large range of tumors, including breast, colorectal, endometrial, and cervical cancers, and glioblastoma [24]. In the kinase domain, the H1047R mutation (exon 20) enhances the linking of p110 to the cell membrane without the requirement of RAS. In the helical domain, the E542K and E545K mutations (exon 9) disturb the interaction with SH2 domains and, consequently, block the action of a regulatory subunit [25,26]. *PIK3CA* mutations promote the resistance to apoptosis, lead to an increased invasive capacity via a possible epithelial-to-mesenchymal transition, induce a chromosomal instability, and favor an immunosuppressive environment [27,28,29,30]. Across all BC subtypes, the prevalence of *PIK3CA* mutations varies between 25 and 40%, the highest being in HR+/HER2− BC [31]. A meta-analysis found a positive association between the *PIK3CA* mutation and ER expression, while for HER2 overexpression, the results are not clear and need more exploration. In TNBC, *PIK3CA* mutations were also associated with the presence of the androgen receptor (AR) [32]. *PIK3CA* mutations are associated with an improvement in recurrence-free survival (RFS) in early BC patients but have no impact on the distant disease-free survival (DFS) or overall survival (OS) [33]. In HR+/HER2− BC, *PIK3CA* mutations do not predict the response to ET at an early stage but the results are unclear in the metastatic setting. A meta-analysis, including 1929 patients mixing all subtypes and early and metastatic stages of BC, found that the *PIK3CA* mutation is an independent poor-prognosis factor [34]. Recently, Mosele et al. found distinct results according to the BC subtypes. HR+/HER2− metastatic BC with the *PIK3CA* mutation appeared to be less sensitive to chemotherapy and had a worse survival than wild-type *PIK3CA*. In TNBC, the data were opposite and the patients with the *PIK3CA* mutation had better survival than patients with wild-type tumors. This may be due to the fact that some TNBCs were initially ER+ tumors that lost ER expression when tumors acquired the *PIK3CA* mutations. It might also be due to AR+ TNBCs, which are enriched in *PIK3CA* mutations and are thought to have a more indolent natural history compared to the other subtypes of TNBC [32]. Several studies showed a different prognostic impact between the two “hotspot” mutations on exon 9 and 20, but the results are unclear and contradictory [35,36,37,38]. In the neo-adjuvant setting, a series of patients (n = 92) with early TNBC treated with an anthracycline-based regimen and *PIK3CA* exon 20 mutation had a lower rate of pathological complete response (pCR); in this trial, exon 20 mutation occurred in only seven patients [39]. Hu et al. also suggested that, in TNBC cell lines and in 50 patients with early TNBC, the *PIK3CA* mutation induced chemoresistance and promoted relapse [40].

An analysis of the *PIK3CA*-mutated cancer genome revealed a co-occurrence of multiple PI3K alterations in approximately 15% of BC cases and these alterations were more frequently mutations in *cis* on the same allele [41]. Multiple pathway alterations lead to hyperactivation compared to a unique alteration, suggesting enhanced oncogenic addiction of double *PIK3CA* mutations and, therefore, a possible increased sensitivity to PI3Kα inhibitors [41]. Activated HER2-HER3 receptors also solicit the PI3K pathway, which suggests that *PIK3CA* mutations may confer some resistance to anti-HER2 therapy [42]. Pre-clinical data found that the *PIK3CA* mutation may confer resistance to trastuzumab and lapatinib [43]. In HER2+ BC, the predictive or prognostic value of *PIK3CA* mutational status was also controversial. In early BC, a study failed to show an impact of *PIK3CA* mutation for trastuzumab sensitivity when administered in the neo-adjuvant or adjuvant setting. Another group conducted by Loibl et al. found that the *PIK3CA* mutation was associated with a significant reduction in pCR in a meta-analysis, including 967 patients treated in the neo-adjuvant setting with trastuzumab or lapatinib. In the metastatic setting, the results are also unclear. In the CLEOPATRA trial (a randomized phase III study in metastatic HER2+ BC receiving trastuzumab plus docetaxel with or without pertuzumab as first-line treatment), Baselga et al. showed that wild-type *PIK3CA* status was associated with a better prognosis, independently from the treatment effect. With lapatinib, a small-molecule HER2 inhibitor, or pertuzumab, another anti-HER2 monoclonal antibody, the *PIK3CA* mutations did not predict a benefit from these therapies [37,44,45,46].

In vitro, the treatment of cell lines by PI3K inhibitors leads to an arrest of proliferation [47]. A direct cytotoxic effect was also reported with a pan-PIK3 inhibitor or a dual inhibitor PI3Kγ-PI3Kα. In the ER+ BC cell line, the appearance of hormone resistance was associated with hyperactivation of the PI3K pathway. The PIK3CA and mTOR inhibitors allow one to induce apoptosis of hormone-independent cells and, consequently, delays the hormone resistance phase [48]. A recent meta-analysis, including eight studies with 2670 patients, showed that the *PIK3CA* mutation is a favorable predictive factor of objective response rate (ORR) (odds ratio: 1.98) and PFS (hazard ratio (HR) = 0.65) in HR+ BC treated by a PI3K inhibitor [49]. 

## 3. Clinical Development of PI3K Inhibitors

### 3.1. Pan-PI3K Inhibitors

These therapies inhibit the kinase activity of all four isoforms of class I PI3K: α, β, γ, and δ. 

#### 3.1.1. Buparlisib (BKM120)

Buparlisib is a 2,6-dimorpholino pyrimidine derivative, which inhibits both PI3K (p110 subunit) and also mTOR, although to a lesser extent [50]. At the preclinical level, it has antitumor activity, preferentially in *PIK3CA*-mutated cell lines, inducing apoptosis in the endocrine-sensitive BC cell lines when combined with estrogen deprivation. Additionally, it demonstrated synergism with trastuzumab in trastuzumab-resistant HER2+ models [51].

Oral buparlisib was first tested in a phase I study involving advanced solid tumors. Approximately 40% of patients experienced a serious adverse event (SAE), including hyperglycemia, skin rash, asthenia, and mood disorder. A dose of 100 mg/day was the recommended dose and a clinical benefit rate (CBR, i.e., objective responses and stable disease for more than 24 weeks) of 41% was observed [52]. A phase I study tested buparlisib in combination with capecitabine in 25 advanced BC patients from all molecular subtypes. The main side effects observed during the study included nausea, hyperglycemia, rash, diarrhea, mucositis, depression, and anxiety. Approximately 50% of patients had a grade ≥3 adverse event (AE). Despite encouraging results with a 68% CBR % [53], the development of buparlisib in association with capecitabine was stopped. In phase Ib studies focusing on HR+/HER2− BC, the 100 mg/day dose was confirmed as safe in association with letrozole [54] or fulvestrant [55], and the observed CBRs were promising. A phase II study examined the efficacy of buparlisib in association with tamoxifen in advanced BC pretreated by ET. The median PFS was 6.1 months but reached 8.7 months in *PIK3CA*-mutated patients. The study was prematurely stopped because of the significant rate of buparlisib-related toxicity [56].

In the BELLE-2 trial, buparlisib was investigated in HR+/HER2− BCs whose disease progressed on or after aromatase inhibitor (AI) use. Patients (n = 1147) received an association of fulvestrant with buparlisib or placebo. Median PFS was significantly but marginally increased in the patients treated with buparlisib (6.9 versus 5.0 months with an HR of 0.78; 95% CI 0.67–0.89, *p* = 0.00021), but the final OS results did not find any significant difference. Regarding the tolerance, 78% of patients receiving the combination experienced grade 3 or 4 adverse events versus 34% in the placebo arm. The main side effects were hyperglycemia, rash, anxiety and some severe mood disturbances (depression and suicide attempts), and elevated ALAT and ASAT [57]. The benefit of buparlisib was similar in patients with PI3K-pathway-activated BC as in the overall cohort. In the BELLE-3 study, buparlisib versus placebo was combined with fulvestrant for HR+/HER2− BC patients (n = 432) progressing on or after prior ET and mTOR inhibitors [58]. The toxicity profile was similar to the one described in BELLE-2. Again, a numerically modest increase in PFS was noted in the buparlisib-treated patients versus placebo-treated ones (3.9 versus 1.8 months with a HR of 0.67; 95% CI 0.53–0.84, *p* = 0.0003), but no effect on OS was noted. In BELLE-3, the subgroup of *PIK3CA*-mutated BC seemed to derive a higher benefit from buparlisib: 4.7 versus 1.4 months, with a better HR of 0.39 (95% CI 0.23–0.65) than in the overall cohort. Both phase III trials suggested focusing on the search for more selective PI3K inhibitors because the clinical benefit was modest compared to the toxicity profile [58,59]. The BELLE-4, phase II/III study tested the addition of paclitaxel to buparlisib for HER2- BC but did not improve PFS. The median PFS was 8.0 months in the experimental arm versus 9.2 months in the placebo arm, with HR of 1.18 (95% CI 0.82–1.68); the study was stopped due to lack of efficacy [60]. 

A phase II study enrolling 50 TNBC patients did not find any response to buparlisib in this subtype [61]. Buparlisib was also explored in association with trastuzumab in a phase Ib trial enrolling trastuzumab-resistant advanced HER2+ BC. The disease control rate was 75% in the population, including approximately 40% of patients with an activated PI3K pathway. However, SAEs were observed in 67% of patients, including hyperglycemia, rash, and diarrhea. Serious mood disorders occurred for 17% of patients [62]. With an ORR of 10%, this study failed to reach its primary endpoint [63]. Our group evaluated the association of buparlisib and lapatinib in a phase Ib study for trastuzumab-resistant HER2+ advanced BC (n = 24). Buparlisib at 80 mg once a day (QD) and lapatinib at 1000 mg QD were feasible with an expected toxicity profile. The CBR was 29%, and one patient experienced a complete response (CR) (4.2%). The SAEs most frequently observed were diarrhea, rash, liver toxicity, hyperglycemia, and mood disorders [64]. NeoPHOEBE trial, a phase II randomized and double-blind study, explored the association of buparlisib, trastuzumab, and paclitaxel in a neo-adjuvant setting. Patients (n = 50) with HER2+ early BC were included. Only 16% of patients had *PIK3CA* mutations. The study was prematurely stopped due to unacceptable liver toxicity [65]. 

#### 3.1.2. Pictilisib (GDC-0941)

Pictilisib potently and selectively inhibits all PI3K isoforms, although being particularly effective on p110α, with a much lesser effect on mTOR [66]. The first-in-human study of pictilisib enrolled several tumor types and found preliminary signs of antitumor activity as a single agent with nausea, diarrhea, vomiting, and rash being frequent AEs [67]. A BC-specific phase Ib trial was conducted, evaluating pictilisib plus paclitaxel with bevacizumab or trastuzumab, according to the molecular subtype. The response rate ranged from 23 to 53%. In a separate cohort enrolling only HR+/HER2− metastatic BC, and receiving pictilisib in association with letrozole, the ORR was 33.3%. All patients experienced an AE and the rate of SAEs was 72.5%, including neutropenia, rash, peripheral neuropathy, pneumonia, and venous thromboembolism [68].

The randomized, double-blind, placebo-controlled, FERGI trial evaluated pictilisib in combination with fulvestrant in post-menopausal women with an advanced or metastatic HR+/HER2− BC resistant to an aromatase inhibitor (AI). The median PFS was 6.6 months versus 5.1 months in the first part enrolling an unselected population, and 6.5 months versus 5.1 months in the second part enrolling only *PIK3CA*-mutated patients [69]. In the PEGGY study, a multicenter, placebo-controlled, phase II randomized trial enrolling 183 patients with locally recurrent or metastatic BC, the addition of pictilisib to paclitaxel did not improve the PFS at interim analysis and the trial was prematurely stopped. The median PFS was 8.2 months in the pictilisib group versus 7.8 months in the placebo group. Approximately 30 patients had *PIK3CA* mutations in both groups, and in this population, the median PFS was 7.3 months for the pictilisib group and 5.8 months for the placebo group [70].

In the neo-adjuvant setting, pictilisib in association with anastrozole was compared with anastrozole alone in a phase II study, enrolling 75 patients with newly diagnosed operable HR+/HER2− BC. A significantly different decrease in the geometric Ki67 average was observed: 83.8% for pictilisib/anastrozole versus 66.0% for the anastrozole group. Geometric Ki67 was used because of high variability in the determination of Ki67 and the approximate lognormal distribution of the data. Ki67 on day 15 was expressed as geometric mean proportions of the baseline and transformed into mean suppression (defined as one minus the geometric means of the proportional changes) [71]. In a subsequent study focusing on a similar population, the ratio of geometric Ki67 average suppression was 0.48 for *PIK3CA* helical domain mutations, 0.63 for wild type, and 1.17 for kinase domain mutations. These results suggested that early BC with helical domain mutations had a poor response to anastrozole monotherapy (mean Ki67 suppression 53.9%) and that it could be increased by the addition of pictilisib (mean Ki-67 suppression 78.1%). Conversely, BC with *PIK3CA* kinase domain mutations responded well to anastrozole and did not seem to benefit from pictilisib [72]. No data are available in subtypes other than HR+/HER2−, and given the disappointing results in this subtype, the clinical development of pictilisib was halted.

#### 3.1.3. Copanlisib (BAY 80-6946)

Although displaying activity against all isoforms of PI3K, copanlisib has a higher inhibitory effect on p110α than on other isoforms and demonstrates predominant antitumor activity in *PIK3CA*-mutated cell lines [73]. A first-in-human study of intravenous copanlisib was conducted in patients with advanced solid cancer. In this study, 16 patients had BC and two of them experienced a partial response (PR). Nausea and hyperglycemia were the main toxicities [74]. A phase Ib trial in patients with advanced solid tumor showed no benefit for an association of weekly copanlisib plus refametinib (MEK inhibitor). In this trial, only four patients had BC and no objective response was observed. Grade 3/4 toxicity occurred in 44 patients (68.8%) with mainly hypertension, diarrhea, and rash [75]. 

A phase Ib/II study explored the association of copanlisib at 45 or 60 mg intravenous (IV) with weekly trastuzumab in patients with metastatic HER2+ BC who progressed after prior anti-HER2 therapy. Most adverse events were hyperglycemia, fatigue, nausea, and hypertension. Stable disease (SD) occurred in six patients (50%) with no objective response. In the phase II part (n = 20), the CBR was 30% and the duration of response was 15.0 weeks. No other clinical result is available with copanlisib in the other BC subtypes, and the clinical development of this drug has been abandoned for solid tumors, whereas it is FDA-registered in adult patients with relapsed follicular lymphoma who have received at least two prior systemic therapies.

### 3.2. PI3Kα-Specific Inhibitors

PI3Kα-specific inhibitors are a group of selective oral inhibitors of the PI3K catalytic subunit p110α class I. Other subunits can be inhibited, but all members of this class have a significant decrease in the effect on PI3Kβ to limit the risk of side effects in common [76].

#### 3.2.1. Alpelisib (BYL719)

With a half-maximum inhibitory concentration in vitro on p110α less than 5 nM, compared to more than 1150 on p110β, alpelisib has demonstrated major antitumor activity in various preclinical models of cancer, notably when associated with *PIK3CA* alterations [77]. A first-in-human study determined the maximal tolerated dose (MTD) of oral alpelisib at 400 mg QD or 150 mg twice a day. Objective tumor responses were observed at doses superior or equal to 270 mg QD, with at least two cycles of experimental treatment, while SD occurred at 180 mg QD. Patients with wild-type *PIK3CA* had no clinical benefit. The most common toxicities were hyperglycemia, diarrhea, nausea, decreased appetite, and vomiting [78]. Baselga et al. explored alpelisib plus everolimus in patients with advanced solid tumor, and alpelisib plus everolimus plus exemestane for post-menopausal patients with advanced HR+ BC. For the triple association, the MTD was 200 mg for alpelisib, 2.5 mg for everolimus, and 25 mg for exemestane, QD. Main SAEs were fatigue and transaminitis. In a BC-specific dose-expansion cohort (n = 11), CBR was 50%, including 25% of PR [79,80]. Alpelisib was also tested in association with LSZ102, a selective estrogen receptor degrader (SERD), in patients with prior ET failure. Independently of the *PIK3CA*- or *ESR1*-mutational status, the median PFS and CBR were 1.8 months and 1.3% for LSZ102 alone and 3.5 months and 20.9% for LSZ102/alpelisib, respectively. In the LSZ102/alpelisib group, patients experienced diarrhea, nausea, hyperglycemia, vomiting, anemia, rash, and transaminitis [81]. A phase Ib study tested alpelisib in combination with letrozole in post-menopausal patients with metastatic HR+/HER2− BC and endocrine-resistant disease. In combination with letrozole, the MTD of alpelisib was 300 mg QD. Side effects were similar to those observed in previous phase I studies. In this trial, CBR was 44% for the *PIK3CA*-mutated patients and 20% for the wild-type *PIK3CA* patients, and was it independent from the presence of *FGFR1/2* amplification and *KRAS* and *TP53* mutations [82]. Another phase Ib study focused on the same population, but alpelisib was associated with fulvestrant. The association was safe and found the same recommended dose of 300 mg. The median PFS was 9.1 months versus 4.7 months and the objective response rate was 29% versus 0% for patients with *PIK3CA*-mutated BC versus those with PIK3CA wild BC, respectively [83]. A phase Ib in premenopausal patients with advanced HR+/HER2− BC evaluated the MTD and preliminary efficacy of tamoxifen plus goserelin, with or without alpelisib. MTD was 350 mg QD and the median PFS was 25.2 months. All patients experienced adverse events, including rash, weight loss, stomatitis, nausea, hyperglycemia, alopecia, and diarrhea. Grade 3/4 adverse events were found in 50% of patients [84]. 

SOLAR-1 study was a randomized, phase III trial enrolling patients with HR+/HER2− advanced BC and disease progression after a prior AI. It compared alpelisib versus placebo in combination with fulvestrant. In the cohort of patients with *PIK3CA*-mutated BC, 169 patients were treated with alpelisib and 172 with placebo. In this population, the median PFS was significantly higher in the alpelisib group compared to the placebo group (11 months versus 5.7 months, HR: 0.65, 95% CI, 0.50 to 0.85, *p* < 0.001). In the cohort of patients without *PIK3CA* mutation, no significant difference was found. Grade 3/4 adverse events occurred in 76% of the cases in the alpelisib group versus 35.5% in the placebo group. The most common side effects were hyperglycemia (63.7%), diarrhea (57.7%), nausea (44.7%), and rash (35.6%) [85]. An anti-histaminic preventive treatment decreased the incidence of rash (26.7% versus 64.1%) [86]. Quality of life (QoL) was also assessed in the SOLAR-1 trial, and no significant difference in the functional status was found for alpelisib versus placebo, with an overall change from baseline at −3.50 versus 0.27, respectively. The main differences concerned diarrhea, anorexia, nausea, and asthenia. Median time to reach 10% of deterioration for QoL was not statistically different between the two arms [87]. The final analysis concerning OS showed that the median OS was 39.3 months in the alpelisib group and 31.4 in the placebo group (HR 0.86, 95% CI 0.64–1.15, *p* = 0.15). Thus, the OS results did not reach statistical significance. For patients with lung and/or liver metastases, the median OS was 37.2 months and 22.8 months, and the median time to chemotherapy was 23.3 months and 14.8 months in the alpelisib and placebo groups, respectively. 

The SOLAR-1 results led the FDA to approve “alpelisib in combination with fulvestrant for post-menopausal women, and men, with HR+/HER2− *PIK3CA*-mutated advanced or metastatic BC following progression on or after an endocrine-based regimen”, whereas the EMA approved alpelisib in a similar population but only “after an hormone treatment used alone has failed”. In SOLAR-1, only 5 to 7% of patients had received a prior CDK 4/6 inhibitor-based ET, which has become the standard of care in this population, making per-label use according to EMA registration very unlikely. Rugo et al. conducted BYLieve, a phase II multicenter, non-comparative study, for advanced HR+/HER2− BC treated with alpelisib in association with fulvestrant after previous administration of CDK 4/6 inhibitors. In this trial, the median PFS was 7.3 months, and the toxicity rate of grade 3/4 was similar to SOLAR-1, with 67% of patients facing adverse events [88]. Results were similar in cohort A, including patients with prior CDK 4/6 inhibitors plus AI, and cohort B, including patients with prior CDK 4/6 inhibitors plus fulvestrant [89]. Similar activity was found in cohorts A and B, whether patients received less or more than 6 months of treatment with a CDK4/6 inhibitor [90,91]. Bello et al. examined the data from the French early-access program of alpelisib in combination with fulvestrant in heavily pre-treated HR+/HER2− advanced BC, including CDK4/6 inhibitors, in a real-life setting (median of four previous lines of systemic treatments for advanced disease). They found a median PFS of 5.3 months (95% CI, 4.7–6.0) and a 6-month CBR of 45.3% (95% CI, 37.8–52.8), but nearly 40% of patients discontinued alpelisib due to adverse events [92].

The SAFIR02-BREAST and SAFIR-PI3K phase II randomized studies enrolled 1462 patients with metastatic HER2− BC and evaluated the maintenance with targeted therapies (including alpelisib with fulvestrant), guided by genomics versus chemotherapy. Patients with endocrine-resistant disease had a *PIK3CA*-mutated advanced BC, which did not progress after 6–8 cycles of first- or second-line chemotherapy. The study found a significant longer PFS for the patients with alterations in the ESMO Scale for Clinical Actionability of molecular Targets (ESCAT), equal to I or II. The median PFS was 9.1 months in the maintenance with targeted therapy group versus 2.4 months in the maintenance chemotherapy group (HR 0.41; *p* < 0.001). For all patients, the median PFS of the two groups was not significantly different, with an HR of 0.77 (95% CI: 0.56–1.06, *p* = 0.109) [93].

Letrozole plus alpelisib was compared to letrozole plus placebo in the neoadjuvant setting in the NEO-ORB trial. This was a randomized, phase II, placebo-controlled trial. Post-menopausal patients (n = 257) were included, 127 patients had a *PIK3CA* mutation, and 130 were wild-type *PIK3CA*. The composite primary endpoint (ORR and pCR) was not met. In the patients with *PIK3CA*-mutated BC, the ORR was 43% and 45% in the alpelisib- and placebo-treated groups, respectively, whereas in the patients with wild-type *PIK3CA* BC, the ORR was 63% and 61%. The pCR rates were low and not statistically different in all groups [94]. 

In advanced TNBC, a phase I/II study evaluated Alpelisib in association with Nab-paclitaxel and found an ORR of 59%, including 7% of CR. The median PFS was 8.7 months. Patients with a *PIK3CA* mutation had a median PFS of 11.9 months versus 7.5 months for patients with Wild-type *PIK3CA* status. They also found a higher median PFS in patients without pre-diabetic or diabetic metabolic status, with 11.9 versus 7.5 months [95]. In the neoadjuvant setting, a phase II trial is evaluating the association of alpelisib with nab-paclitaxel in patients with early refractory TNBC or with suboptimal response to initial anthracycline-based chemotherapy and displaying *PIK3CA* or *PTEN* alterations [96]. Recently, a phase I trial evaluated the combination of alpelisib and olaparib in pretreated advanced TNBC or any subtype with germline *BRCA*-mutated BC. The hypothesis was that PI3K inhibition could induce a functional homologous recombination deficiency (HRD), leading to higher sensitivity to PARP inhibitors. The recommended phase II dose was a reduced posology of both agents (200 mg olaparib, 200 mg alpelisib). The main toxicities were hyperglycemia, rash, fatigue, anorexia, and nausea. The ORR and CBR were 18% and 35%, respectively. Importantly, none of the responders had *BRCA* mutation or any other HRD gene mutation, as well any alteration in the PI3K pathway, consistent with a functional synergism between the two drugs [97]. 

A phase I trial examined the combination of alpelisib with trastuzumab emtansine (TDM-1) in patients with pretreated HER2+ metastatic BC. No unexpected signal of toxicity was seen and the ORR and CBR were 43% and 71%, respectively, including 30% and 60% in patients with prior resistance toTDM-1. The median PFS was 8.1 months, consistent with the central position of the PI3K pathway in the mechanisms of resistance to anti-HER2 treatment [98]. Another phase I study, including HER2+ BC with *PIK3CA* mutations, investigated alpelisib in combination with trastuzumab and LJM716 (an HER3-targeted antibody). The study revealed significant gastrointestinal toxicity for the triple association and was stopped prematurely [99]. A phase III, multicenter, randomized, double-blind, placebo-controlled trial, evaluating maintenance Alpelisib in association with trastuzumab and pertuzumab for metastatic HER2+ BC with *PIK3CA* mutation, is ongoing [100].

#### 3.2.2. Taselisib (GDC-0032)

Taselisib inhibits tumor cell proliferation and induces apoptosis by potently and selectively inhibiting p110α, particularly in the context of activating *PIK3CA* mutation. Of note, its mechanisms of action also include a proteasome-mediated degradation specific to the mutant oncoprotein [101]. A phase I trial of taselisib was conducted enrolling patients with advanced solid tumors. The side effects were similar to those of alpelisib, including diarrhea, hyperglycemia, decreased appetite, nausea, rash, stomatitis, and vomiting. The ORR was 36% in the subset of BC with *PIK3CA* mutations, while no response was observed in patients with wild-type *PIK3CA* tumors [102]. In an additional large phase I trial of 166 patients with *PIK3CA*-mutated cancer, the ORR was 9%. In a subset of 17 patients with advanced TNBC, the ORR was 5.9%. Nevertheless, the response rate seemed to be higher in the case of helical domain mutations. Mutations in *TP53* and *PTEN* seemed to be associated with resistance to taselisib [103]. A phase Ib study explored taselisib in combination with tamoxifen in patients with HR+ metastatic BC whose disease progressed after at least one prior line of ET (n = 30). The frequently observed adverse events were diarrhea, mucositis, and hyperglycemia. The ORR was 24%, and 40% of patients had an SD at 6 months or more [104]. A phase Ib dose-escalation and dose-expansion trial evaluated the association of taselisib plus taxane in metastatic BC and non-small-cell lung cancer (NSCLC). Seventy patients had BC, seven had NSCLC, and one had both. SAEs occurred in 90.5% (fatigue, diarrhea, nausea, and neutropenia), with 14.5% of adverse events leading to death. The ORR was 35% in the docetaxel arm and 20.4% in the paclitaxel arm. Despite interesting activity signals, such major toxicity limited further development of these associations [105].

A phase Ib trial enrolling heavily pretreated advanced HR+/HER2− BC patients evaluated taselisib with letrozole (n = 28). The ORR was 38% in the *PIK3CA*-mutated group and 9% in the wild-type *PIK3CA* group. The patients experienced diarrhea, hyperglycemia, and mucosal inflammation grade 3 or 4 adverse events, in 14%, 7%, and 7%, respectively [106]. The LORELEI trial was a phase II, multicenter, randomized, double-blind, placebo-controlled study of letrozole in association with taselisib versus letrozole, with placebo for early HR+/HER2− BC in the neoadjuvant setting (n = 334), 46% of them displaying *PIK3CA* mutations. The ORR was 50% in the taselisib group versus 39% in the placebo group (*p* = 0.049), but the pCR rate was very low in both arms (1 to 2%). An AE ≥ grade 3 occurred in 12% of patients, mainly with digestive or infectious toxicity [107]. Taselisib was also investigated in combination with fulvestrant in post-menopausal women with advanced HR+/HER2− BC, in a non-comparative phase II study enrolling 87 patients. For patients with *PIK3CA*-mutated BC, the ORR and CBR were 29.5% and 38.5%, respectively. Approximately 50% of patients experienced a grade 3 or more AE, which was colitis in 13% of cases. Of note, 21% of patients harbored both *ESR1* and *PIK3CA* mutations [108,109]. These results led to a phase III study, the SANDPIPER trial, for *PIK3CA*-mutant advanced HR+/HER2− BC resistant to ET, evaluating fulvestrant associated with taselisib (n = 430) versus placebo (n = 176). The toxicity was significant, with 49.5% of patients experiencing an SAE in the taselisib arm versus 16.4% in the placebo arm. The most frequent AEs were diarrhea, hyperglycemia, rash, stomatitis, and colitis. A statistically significant but relatively limited improvement in PFS (7.4 months versus 5.4 months, HR 0.70, *p* = 0.0037) was observed in the taselisib-treated patients. The ORR was also increased with taselisib to 28.0% versus 11.9% with the placebo [110]. A multicenter, randomized phase II trial, enrolling patients with metastatic BC whose disease progressed after at least one prior line of ET, evaluated tamoxifen in association with taselisib or placebo. The median PFS was 3.2 months in the placebo arm and 4.8 months in the taselisib arm (unstratified HR 0.62, 95% CI 0.43–0.93) and CBR was 14.5% versus 22.4%. Approximately 24% of patients harbored a *PIK3CA* mutation. The toxicity rate of grade ≥ 3 adverse events was 44% in the taselisib arm versus 5% in the placebo arm, with no new safety signal [111]. 

Lehmann et al. conducted a phase Ib/II study in patients with metastatic AR+ BC to evaluate enzalutamide in combination with taselisib or placebo. Phase I included patients with HR+ BC and TNBC, whereas phase II included only patients with TNBC. In TNBC, the CBR was 35.7% in the taselisib arm versus 0% in the placebo arm. The CBR was 42.9% in *PIK3CA*-mutated patients versus 28.6% in wild-type *PIK3CA*. The CBR was 75% in the luminal androgen receptor (LAR) subtype versus 12.5% in the other subtypes (*p* = 0.06) [112]. In HER2+ BC, a phase Ib trial evaluated taselisib in combination with T-DM1 in metastatic BC, including 23% displaying *PIK3CA* mutations. Grade 3–4 toxicities included pneumonitis and thrombocytopenia at rates of 15.3% and 19.2%, respectively. The median PFS was 7.6 months and ORR was 33%, ranging from 40% to 22%, whether *PIK3CA* mutations were present or not, respectively [113].

Table 1 summarizes the phase III trials with PI3K inhibitors in advanced BC.

#### 3.2.3. Inavolisib (GDC-0077)

Inavolisib is a recently developed, orally administered, strong p110α inhibitor, which induces a specific degradation of the mutated form of PIK3CA, as already described with taselisib, associated with an exquisite activity in *PIK3CA*-mutated cancer preclinical models [114,115]. A first-in-human, phase I dose-escalation, multi-arm trial of inavolisib included patients with advanced or metastatic tumors harboring *PIK3CA* mutations. The main significant toxicities included hyperglycemia, lymphopenia, fatigue, nausea, and weight loss. The CBR was estimated at 45%, with approximately 20% of patients with PR [116]. This phase I trial also examined the association of inavolisib with letrozole, with or without palbociclib. No DLT was observed at the recommended dose of inavolisib, which was 9 mg QD in both arms. The major SAEs were hyperglycemia, fatigue, hypokalemia, hyperglycemia and neutropenia, leukopenia, and thrombocytopenia in patients receiving inavolisib, without or with palbociclib, respectively. The CBR without or with palbociclib was 35% and 76%, respectively. A pharmacokinetic exploration did not find any interaction between the three drugs, and a pharmacodynamics study found an important downregulation of the PI3K pathway via a reduction in phosphorylated AKT [116]. 

In a phase Ib trial, the association of inavolisib with fulvestrant was examined in post-menopausal women with *PIK3CA*-mutated metastatic BC. The toxicity profile was similar to those previously described, except for a higher rate of hyperamylasemia and hyperlipasemia. The CBR was 60%, including 36% of patients achieving PR [117]. The association of inavolisib with fulvestrant and palbociclib was also examined in a phase Ib study enrolling *PIK3CA*-mutated HR+/HER2− metastatic BC patients. The grade ≥ 3 toxicities included hyperglycemia (47%) and neutropenia (47%). The CBR was evaluated at 61%, and 28% of patients had a PR [118]. 

#### 3.2.4. Serabelisib (TAK-117)

Serabelisib is a novel PI3K inhibitor, with a very high level of selectivity against p110α and a strong ability to induce an inhibition of cell proliferation and apoptosis. A first-in-human phase I trial of oral serabelisib enrolled patients with advanced solid tumors and tested continuous or discontinuous regimens. The rate of grade ≥ 3 AEs ranged from 22% to 35%, depending on the drug regimen. The most frequent adverse events were ALAT/ASAT elevation and hyperglycemia. The response rate was low, and the use of serabelisib as a single agent required an intermittent schedule. Thus, given the limited efficacy, the development in monotherapy was stopped [119]. Nevertheless, the association of TAK-228 (oral mTORC1/2 inhibitor) with serabelisib was examined in metastatic TNBC, with the hypothesis that it could increase the tumor genomic instability and, consequently, increased the TNBC sensitivity to chemotherapy and immunotherapy. In a small-sized study of 10 patients with pre-treated TNBC, one PR, three SDs and six progressive diseases (PDs) were observed. Interestingly, a durable response to pembrolizumab after progression under this regimen was noted in 3 out of 10 patients [120]. The association of serabelisib-TAK-228 was also tested with paclitaxel, in patients with advanced ovarian, endometrial, or BC. The ORR was 47% in all cohorts, and the median PFS was approximately 11 months. The proportion of drug-related grade ≥ 3 adverse events was 9% [121]. 

Table 2 summarizes the ongoing trials of PI3K inhibitors in BC.

## 4. PI3K Inhibitors or CDK4/6 Inhibitors or Both in HR+/HER2− Disease?

While accumulating data have shown a survival benefit for adding a CDK4/6 inhibitor to ET in endocrine-resistant disease, no clinical trial has been performed comparing this class of drugs versus therapies targeting the PI3K/Akt/mTOR pathway, including PI3K inhibitors, when associated with ET. A recent study conducted by Han et al. used a network analysis, allowing for an indirect comparison between these two approaches, and it found an improvement in median PFS in favor of CDK4/6 inhibitors, as compared with PI3K inhibitors. This study included randomized trials evaluating the benefit and toxicity of CDK4/6 inhibitors or PI3K/AKT/mTOR inhibitors with ET between 2010 and 2019. A large cohort of 9771 patients was analyzed by a generic inverse variance method, allowing them to indirectly estimate the pooled HR and using a Bayesian framework to validate models with Markov Chain Monte Carlo methods. PFS was superior in the CDK4/6 inhibitors plus ET group compared to the PI3K inhibitors group (HR, 0.73; 95% CI, 0.62–0.86), while no statistical difference was shown in the AKT and mTOR inhibitors group. The benefits were observed in all subgroups, all metastatic lines, and in both visceral and bone-only diseases. The benefit of CDK4/6 inhibitors was also observed in OS, with a pooled HR at 0.78 (95% CI, 0.65–0.94) when they were compared with the PI3K and mTOR inhibitors. A risk ratio (RR) was used to evaluate grade ≥ 3 AEs. Cytopenia had a higher rate with CDK4/6 inhibitors. For cytopenia, the RR was significantly different in favor of abemaciclib versus palbociclib (RR = 0.024) and ribociclib (RR = 0.018). Hyperglycemia had a higher rate with the PI3K and mTOR inhibitors. RR was similar in the CDK4/6 and PI3K/AKT/mTOR inhibitors groups [122]. A second meta-analysis identified 79 phase II/III studies, including post-menopausal women, with HR+/HER2− BC, whose disease progressed after first-line treatment with AI or ET, and explored the CDK4/6 inhibitors or PI3K/AKT/mTOR inhibitors in association with fulvestrant. For the analysis, only eight studies were included. The authors showed that the CDK4/6 inhibitors (abemaciclib, palbociclib, and ribociclib) were superior in PFS, ORR, and OS compared to the PI3K inhibitors (pictilisib, alpelisib, and buparlisib) [123].

Robust pre-clinical data suggested a synergistic effect to the combination of CDK4/6 inhibitors with PI3K inhibitors (buparlisib and alpelisib) in HR+/HER2− BC. In a mouse model, O’Brien et al. demonstrated that the association of a CDK4/6 inhibitor to an ET significantly inhibited tumor growth compared to ET alone. Moreover, PI3K inhibitors in addition to CDK4/6 inhibitors and ET induced a tumor regression and one CR was also observed [124]. In an in vitro and in vivo study of TNBC cell lines with *PIK3CA* mutations, Asghar et al. found a synergistic effect of CDK4/6 inhibitors with PIK3CA inhibitors [125].

Concerning HR+/HER2− metastatic BC, a phase Ib trial tested the safety and preliminary efficiency of the association of ribociclib (continuous versus discontinuous) and fulvestrant, with/without alpelisib or buparlisib. Both arms with triple associations were suspended due to unacceptable toxicity and were not continued in phase II investigations [126]. Alpelisib was also explored in association with ribociclib and letrozole in a phase Ib/II trial dedicated to post-menopausal women. Three arms were tested, A1: letrozole/ribociclib (n = 41), A2: letrozole/alpelisib (n = 21), and A3: letrozole/ribociclib/alpelisib (n = 36). In the A3 patients, 22% had SD and 19% had PD. The safety profile was acceptable with the triple association, with grade 3/4 adverse events occurring in 22% for neutropenia, 17% for hyperglycemia, 11% for fatigue, and 6% for nausea. Arm A3 showed a considerable reduction in the Ki76 staining on tumor tissue compared to the A1 or A2 arms [127]. These results must be validated in a large, randomized trial. The triple association of palbociclib, taselisib, and fulvestrant was also explored in a phase Ib trial, conducted by Pascual et al., in patients with *PIK3CA*-mutated, HR+/HER2− advanced BC. In this population, with a median of three prior systemic therapy lines for advanced disease, the ORR was 37.5%. The high baseline cyclin E1 expression and early detection of ctDNA were associated with a shorter PFS. With the triple association, the most frequent grade 3/4 AEs were neutropenia (47.4%) and leucopenia (16.7%) [128]. 

Concerning the association of CDK4/6 inhibitors and PIK3CA inhibitors in TNBC, no trial was published. A pre-clinical study showed a synergistic effect, including a complete and durable response to combined CDK4/6 and PIK3CA inhibitors, in a mouse model [129]. For HER2+ BC, several preclinical studies demonstrated a favorable impact of CDK4/6 inhibitors and PIK3CA inhibitors, but their association was not evaluated at the clinical level [130]. 

## 5. Resistance to PI3K Inhibitors

Despite the central role in tumorigenesis of PI3K, only a modest benefit of PIK3CA inhibitors was observed in the clinical trials for both pan-PIK3CA and PIK3CAα-specific inhibitors. Thus, the clinical benefit could be limited by several resistance mechanisms. Figure 2 summarizes the main mechanisms of resistance to PI3K inhibitors and the corresponding therapeutic strategies to overcome them.

In BC and in prostate cancers, the role of the proviral integration of Moloney virus (PIM) kinase family has been highlighted. Song et al. found that PIM1 kinase promotes resistance and prevents cell death by decreasing the cellular level of ROS, which consequently attenuates the toxicity of PI3K inhibitors [131]. Furthermore, PIM1 conferred an acquired resistance to the PI3K inhibitors, since it appeared that PIM1 overexpression and *PIK3CA* mutations were mutually exclusive in treatment-naïve BC. In tumor biopsies obtained on disease progression under PI3K inhibitors, PIM1 was found to be overexpressed [132]. Based on the above-mentioned data, IBL-302, a PIM/PI3K/mTOR inhibitor, was developed. Pre-clinical data in BC cell lines demonstrated an antitumor efficacy in monotherapy and in combination with trastuzumab for HER2+ BC cell lines resistant to trastuzumab [133]. In BC cell lines, Litchfield et al. reported that abemaciclib can inhibit mTOR by blocking both PIM kinases and CDK4/6, which resulted in blocking the PI3K/Akt/mTOR pathway and reduced cell growth. When abemaciclib was associated with alpelisib in *PIK3CA*-mutated HR+ BC, it had a synergistic effect in neutralizing mTOR activation and increasing the effects on cellular proliferation. In fact, mTOR may be activated by three different pathways that are independent of each other, Akt via PI3K, PIM kinase, and CDK4. These results suggested a potential novel approach to counteract the acquired resistance mechanisms to PI3K inhibitors [134]. 

A network-based mathematical model was developed in HR+ *PIK3CA*-mutated BC cell lines, to identify the determinants of sensitivity or resistance to alpelisib. A novel resistance mechanism was proposed, involving Forkhead Box O 3 (FOXO3) downregulation. In addition, this model predicted the efficacy of alpelisib in combination with myeloid cell leukemia-1 (MCL1) inhibitors, a BH3 mimetic, which was further validated in BC cell lines [135]. Under physiological conditions, AKT negatively regulates FOXO3. AKT downregulation by PI3K inhibitors induces an activation of FOXO3 that can, therefore, be drawn into the nucleus and, then, activates transcription factors, leading to upregulation of RTK, such as HER2 or HER3. In BC, SGK3 is significantly higher in HR-positive disease than in TNBC. PI3K/AKT/mTOR pathway inhibition induced upregulation of the serum and glucocorticoid regulated kinase 3 (SGK3), and SGK3 can phosphorylate various substrates with several substrates in common with AKT and can activate mTOR independently of AKT. This is related to a strong homology between the two catalytic subunits [136]. An in vitro study showed, in *PIK3CA*-mutated BC cell lines, that both AKT and ERK were suppressed with pictilisib or alpelisib. Moreover, if RAS mutation co-occurred, the PI3K inhibitor failed to suppress the ERK pathway [137]. In BC cell lines, the upregulation of *RPS6KA2* (RSK3) or *RPS6KA6* (RSK4) promoted resistance to a PI3K inhibitor, and an RSK inhibitor in addition to PI3K inhibitor could overcome the resistance [138]. For patients with *PIK3CA*-mutated head and neck and esophageal squamous cell carcinomas, Elkabets et al. showed a novel acquired resistance mechanism to alpelisib. Patients who progressed after alpelisib presented an overexpression of AXL. AXL activated the epidermal growth factor receptor (EGFR) by dimerization/phosphorylation and then activated protein kinase C that led to mTOR activatione [139].

A higher activity of alpelisib has been associated with complete inhibition of mTORC1, and an increase in mTORC1 activity was found in BC cell lines with *PIK3CA* mutation and disease progression under alpelisib. Accordingly, the association of an mTOR inhibitor to alpelisib seemed to reverse the drug resistance [140]. Cai et al., using a whole-genome screen on HR+ *PIK3CA*-mutated BC cell lines, found that the loss of several genes (*TSC1*, *TSC2*, *TBC1D7*, *AKT1S1*, *STK11*…), which negatively regulate mTORC1, conferred resistance to PI3K inhibitors, and confirmed that mTOR inhibition could overcome resistance. Genomic alterations in mTORC1 regulators were found in 15% of HR+ *PIK3CA*-mutated BC samples, all of them being mutually exclusive [141]. 

Juric et al. analyzed progressive metastatic sites of patients who died after alpelisib treatment. Compared to the first biopsy, all sites harbored *PTEN* loss. Moreover, the *PTEN* alterations were different depending on sites, and all genomic alterations in *PTEN* led to resistance [142]. Razavi et al. examined biological samples from HR+ BC patients treated with alpelisib and AI in a phase I/II trial and searched for mechanisms of resistance. *PTEN* loss and *ESR1* mutation were associated with resistance [143]. *PTEN* alterations (either mutation or homozygous deletions) could be found at baseline, associated with rapid disease progression, but also in the post-treatment samples, consistent with an acquired mechanism of resistance. Of note, *PTEN*-deficient cells are dependent on p110β for maintaining downstream AKT signaling and, thus, could be more sensitive to PI3Kβ inhibition or a combination of PI3Kα and PI3Kβ [144,145].

Hyperglycemia was one of the most frequent AEs found with PIK3CA inhibitors, with rates equal to 64% in SOLAR-1 [85], 43% in BELLE-2 [57], 37% in BELLE-3 [58], and 40% in the SANDPIPER trial [110]. For patients in which alpelisib was stopped due to unacceptable toxicity, hyperglycemia represented approximately 6%. Importantly, such hyperglycemia might induce plasma insulin rebound, thus leading to restoration of PI3K signaling and, ultimately, to treatment resistance [146]. In addition to the impact on the metabolism, an in vivo study showed that hyperglycemia could influence the BC tumor microenvironment. In co-culture in hyperglycemia compared to normal glycemia, adipocytes produced more inflammatory cytokines (IL-6, IL-8) and growth factors (VEGF, IGF1) and induced a more aggressive and invasive phenotype for BC cells in zebrafish [147]. An ongoing phase II study, entitled “Alpelisib, Fulvestrant and Dapagliflozin (SGLT2 inhibitor) for the Treatment of ER-positive, HER2-negative, *PIK3CA*-Mutant Metastatic Breast Cancer”, includes the same population as the SOLAR-1 trial to establish if the addition of dapagliflozin, an SGLT-2 inhibitor, could significantly reduce hyperglycemia of any grade (NCT05025735). Another phase II study in the same population examined if the association of dapagliflozin and metformin could reduce grade 3 hyperglycemia (EPIK-B4/NCT04899349). A Phase Ib/II study of serabelisib in combination with canagliflozin (an SGLT-2 inhibitor) in advanced solid tumors with *PIK3CA* mutations or *KRAS* mutations, including BC, is testing the efficiency of this association (NCT04073680). A phase II trial, “Preventing High Blood Sugar in People Being Treated for Metastatic Breast Cancer”, with three experimental arms in a similar population as SOLAR-1,a is exploring the rate of hyperglycemia grade 3 a with ketogenic diet, low-carbohydrate diet, or canagliflozin in combination with alpelisib and fulvestrant (NCT05090358).

Some of the adaptive mechanisms of resistance to PI3K inhibitors are described here (shown in yellow in the figure), along with their consequences (shown with red dotted arrows) and potential therapeutic possibilities (shown with green arrows). Loss of the tumor suppressor phosphatase and tensin homolog (PTEN) can appear before or after PI3K inhibitor use and lead to phosphatidylinositol 3,4,5 triphosphate (PIP3) overexpression. All PI3K inhibitors induce a disruption in the carbohydrate metabolism with hyperglycemia; hyperglycemia brings feedback that leads to insulin upregulation and reactivation of tyrosine kinase receptors (RTKs). Activation of mTOR, independent of protein kinase B (AKT), represents an important part of acquired resistance to PI3K inhibitors. We can find many paths of resistance, including the serum and glucocorticoid-regulated kinase 3 (SGK3), the proviral integration of Moloney virus (PIM), or PKC via activation of AXL and RTKs; these molecules induce activation of mTOR and, therefore, the pathway. AKT inactivation induces hypophosphorylation of Forkhead Box O 3 (FOXO3) and allows for the nuclear localization of FOXO3. FOXO3 induces myeloid cell leukemia-1 (MCL1) activation and upregulation of RTKs, leading to survival and overexpression of the PI3K pathway, respectively. In the RAS/MEK/ERK pathway, the PI3K inhibitor can inhibit ERK; however, the appearance of the *KRAS* mutation can induce activation of the pathway, despite ERK inhibition, and it has been observed that PI3K inhibitors induce upregulation of RPS6KA2 (RSK3) and RPS6KA6 (RSK4). L: ligand; IR: insulin receptor; PKC: protein kinase C; G: glucose; I: insulin.

## 6. Conclusions

PI3K inhibitors, including pan-inhibitors and selective inhibitors, are at the beginning of their development in BC. Although they have led to few authorizations so far, their field of evolution is considerable with the development of new more selective inhibitors. Pan-inhibitors have not shown the expected efficacy, but they have opened the way to PI3Kα-specific inhibitors. All BC subtypes have their own metabolic pathway, and a better understanding of specific mechanisms of resistance to these drugs in each subtype is necessary to adapt our therapeutic strategies. Despite the central role of PI3K in oncogenesis, only a modest antitumor activity has been observed, and the future of these drugs rests on the right choice of treatment association to limit the appearance of resistance via another molecular pathway. However, this is a toxicity profile that should not be overlooked, and learning to manage these drugs to facilitate their use as well as discovering predictive biomarkers of responses are the new challenges we must meet.

## Figures and Tables

**Figure 1 cancers-15-01416-f001:**
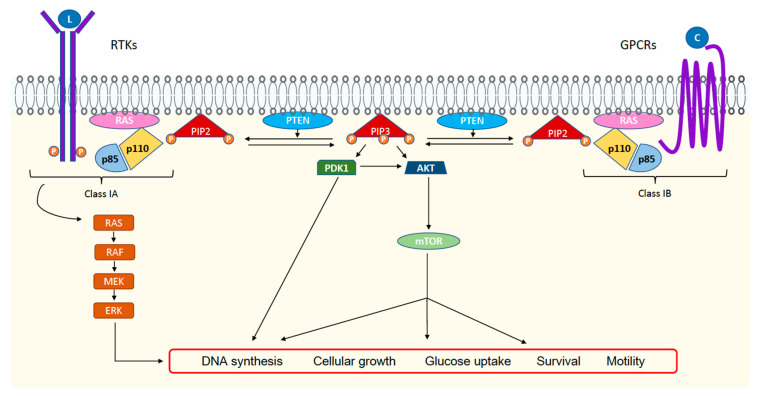
Summary diagram of the PI3K pathway and cellular activation pathways.

**Figure 2 cancers-15-01416-f002:**
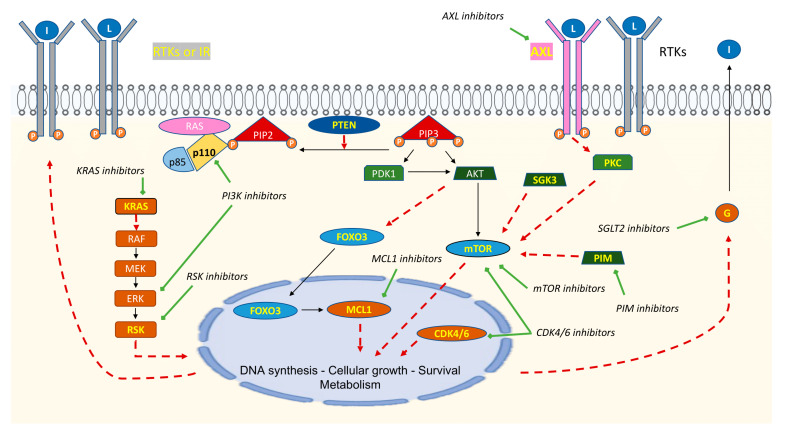
Summary diagram of main mechanisms of resistance to PI3K inhibitors and potential inhibitors to counteract.

**Table 1 cancers-15-01416-t001:** Phase III clinical trials of main PI3K inhibitors in breast cancer and approvals.

Molecule	PIK3CA Subunit Target	Trial (Reference)	Number of Patients	Patients	Drugs	Results	Approval (FDA/EMA)
Buparlisib	Pan PI3K Inhibitor	BELLE-2 [59]	1147	HR+/HER2− ABC progressed on or after AI and a maximum of one previous line of CT	Fulvestrant + Buparlisib or Placebo	mPFS 6.9 vs. 5.0 months HR 0.78 (95% CI 0.67–0.89, *p* = 0·00021)	No approval
		BELLE-3 [58]	432	HR+/HER2− ABC progressed on or after prior ET and mTOR inhibitors	Fulvestrant + Buparlisib or Placebo	mPFS 3.9 vs. 1.8 months HR 0.67 (95% CI 0.53–0.84, *p* = 0·00030)	
		BELLE-4 [60]	416	HR+/HER2− ABC receiving first-line CT	Paclitaxel + Buparlisib or Placebo	mPFS 8.0 vs. 9.2 months HR 1.18 (95% CI 0.82–1.68)	
Alpelisib	p110α selective PI3K inhibitor	SOLAR-1 [85]	341	HR+/HER2− ABC with PIK3CA mutation and disease progression on or after prior AI	Fulvestrant + Alpelisib or Placebo	mPFS 11.0 vs. 5.7 months HR 0.65 (95% CI, 0.50 to 0.85; *p* < 0.001)	* FDA: for HR+ HER2− ABC PI3K-mutated who had received ET previously * EMA: for HR+ HER2− ABC PI3K-mutated who had received ET alone previously
Taselisib	Dual p110α/δ selective PI3K inhibitor	SANDPIPER [110]	631	HR+/HER2− ABC PIK3CA mutated resistant to ET	Fulvestrant + Taselisib or Placebo	mPFS 7.4 vs. 5.4 months HR 0.70 (95% CI, 0.56–0.89 *p* = 0.0037)	No approval

FDA: food and drug administration; EMA: European medicine agency; HR: hazard ratio; HR: hormone receptor; HER2: human epidermal receptor 2; ABC: advanced breast cancer; mPFS: median progression-free survival; CI: confidence interval; vs.: versus; AI: aromatase inhibitor; ET: endocrine therapy; CT: chemotherapy. * Definition of the approval.

**Table 2 cancers-15-01416-t002:** Ongoing phase I–III trials of PI3K inhibitors in breast cancer.

Indication: Molecular Subtype	Population	Trial	Phase	Drug	Number of Patients	Primary Endpoint
HR+ HER2−	ABC Postmenopausal Post AI + CDK4/6i ≤1 line of prior CT > or =1 prior line	NCT05038735 / EPIK-B5	III	Fulvestrant + Alpelisib or Placebo	234	PFS
	ABC Postmenopausal Pre/peri-menopausal (if LHRH agonist)	NCT04191499 / INAVO 120	II/III	Palbociclib + Fulvestrant + Inavolisib or Placebo	400	PFS
	ABC Postmenopausal Pre/peri-menopausal (if LHRH agonist)	NCT05646862 / INAVO 121	III	Fulvestrant + Alpelisib or Inavolisib	400	PFS
	ABC	NCT04355520	I/II	Fulvestrant + TQ-B3525 (Selective PI3K α/δ inhibitor)	42	DLT
	ABC	NCT05504213	I	Fulvestrant + HS-10352 (Selective PI3K α inhibitor)	224	MTD MAD ORR
	ABC	NCT03056755	II	Fulvestrant + Letrozole + Goserelin + Leuprolide + Alpelisib	384	PFS 6months
	ABC	NCT05631795 / ALPINIST	IV	Fulvestrant + Alpelisib	100	SAE ADR
	ABC	NCT04856371	I	Fulvestrant + Letrozole + Palbociclib + CYH33 (Selective PI3K α inhibitor)	228	DLT
	BC Ovarian cancer Solid tumor DDR gene mut +/− PIK3CA mut, Progressed prior PARP inhibitor	NCT04586335	I	Olaparib + CYH33 (Selective PI3K α inhibitor)	350	DLT ORR
	ABC/Stage IV	NCT05216432	I	Fulvestrant + RLY-2608 (Selective PI3K α inhibitor)	190	MTD AE SAE
	ABC/Stage IV	NCT05501886	III	Palbociclib + Fulvestrant + Alpelisib or Gedatolisib (Dual PI3K/mTOR Inhibitors)	701	PFS
	Stage IV	NCT03939897	I/II	Abemaciclib + Fulvestrant + Copanlisib	204	DLT PFS
TNBC	ABC or stage IV ≤1 line of prior CT Part A—PIK3CAmut regardless of PTEN Part B1—PIK3CAmut PTEN loss Part B2—PTEN loss without PIK3CAmut	NCT04251533 / EPIK-B3	III	Nab-paclitaxel + Alpelisib or Placebo	137	PFS (A and B2) ORR (B1)
	Stage IV	NCT05660083 / MpBC	II	Nab-paclitaxel + Alpelisib + L-NMMA (iNOS inhibitor)	36	R2PD ORR
	BC Renal cell carcinoma	NCT03961698 / MARIO-3	II	Nab-paclitaxel + Bevacizumab + Atezolizumab + IPI-549 (Selective PI3K γ Inhibitor)	91	CR
	Multi tumor	NCT02637531	I	Nivolumab + IPI-549 (Selective PI3K γ Inhibitor)	219	DLT AE
	Multi tumor	NCT02646748	I	Pembrolizumab + INCB050465 (Selective PI3K δ inhibitor)	159	Safety
	ABC AR+ PTEN positive	NCT03207529	I	Enzalutamide + Alpelisib	28	MTD
HER2+ and/or HR+	Depending on each group	NCT03006172	I	A—Inavo B—Inavo + Palb/Let C—Inavo + Let D—Inavo + Fulv E—Inavo + Palb/Fulv F—Inavo + Palb/Fulv/Met, G—Inavo + Trastu/Pertu/HT	256	DLT R2PD SAE
	Early BC	NCT05306041 / GeparPiPPa	II	Neoadjuvant: PHESGO + HT + Inavolisib	170	pCR
	ABC/stage IV	NCT03767335 / B-PRECISE-01	I	Trastuzumab + MEN1611 (Selective PI3K α inhibitor) +/− Fulvestrant	62	MTD
	ABC	NCT05063786 / ALPHABET	III	Exp: Trastuzumab + Alpelisib +/− Fulvestrant Control: Trastuzumab + Vinorelbine + Capecitabine + Eribulin	300	PFS
HER2+	ABC	NCT04208178 / EPIK-B2	III	Trastuzumab + Pertuzumab + Alpelisib or Placebo	548	DLT PFS
	Stage IV	NCT04108858	I/II	Trastuzumab + Pertuzumab + Copanlisib	12	SAE DLT PFS
	ABC/Stage IV No limit of prior lines	NCT02390427	I	Trastuzumab + Pertuzumab + Trastuzumab emtansine + Paclitaxel + Taselisib	68	MTD
	ABC/stage IV	NCT02705859 / Panther	I	Trastuzumab + Copanlisib	26	MTD CBR
	Stage IV	NCT05230810	I/II	Fulvestrant + Tucatinib + Alpelisib	40	Safety PFS

ABC: advanced breast cancer; HR: hormone receptor; HER2: human epidermal receptor 2; TNBC: triple negative breast cancer; AR: androgen receptor; HT: hormone therapy; PFS: progression-free survival; PFS2: next treatment progression-free survival; OS: overall survival; ORR: objective response rate; CBR: clinical benefice rate; DOR: duration of response; DOCR: duration of complete response; DCR: disease control rate; BOR: best overall response rate; TTR: time to response; TTCR: time to complete response; QoL: quality of life; AUC: Area Under the Concentration Time-Curve; Cmax: Maximum Plasma Concentration; Cmin: Minimum Plasma Concentration; AE: adverse events; SAE: serious adverse events; ADR: Adverse Drug Reactions; DLT: dose limiting toxicities; TTD: time to deterioration; MTD: maximum tolerated dose; RP2D: recommended phase II dose; MAD: Maximum applicable dose; TILs: Tumor Infiltrating Lymphocytes; IHC: immunohistochemistry; PK: pharmacokinetics markers; ctDNA: circulating tumor DNA; TTFST: time to first subsequent therapy; TTF: time to treatment failure; Inavo: Inavolisib; Palb: Palbociclib; Fulv: Fulvestrant; Let: Letrozole; Met: Metformin; Trastu: Trastuzumab; Pertu: Pertuzumab.

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
