# Peer review of "Phosphoinositide 3-Kinase (PI3K) Inhibitors and Breast Cancer: An Overview of Current Achievements"

_cancers, 2023, doi:10.3390/cancers15051416_

Round 1

Reviewer 1 Report

The authors present a review of PI3K inhibitors both pan-inhibitors and more specific sub-unit inhibitors. The review presents data concerning the fundamental associations between the PI3K molecular pathway and various cancers including breast cancer. The authors do an excellent job of then going on to discuss the state of the art in clinical deployment of PI3K inhibitors both in clinical practice and clinical trials. The review is timely, readable and comprehensive. Where appropriate, the authors also include conclusions and speculation around mechanisms of action, in particular around putative mechanisms of resistance, within the PI3K pathway.

Although overall an excellent review and eminently readable, there are numerous minor errors in syntax which detract from the review and require amendment. In particular the words “the” and “a” or “an” are frequently left out where they should be used. The instances of this are too numerous to point out specifically and the review would benefit from correction in this area throughout the manuscript.

The review could also benefit from the inclusion of a further figure indicating potential resistance mechanisms and how they could arise post-therapy in particular.

Other minor comments:

Line 105: Please change “with at the top of the list PIK3CA and PTEN” to “with PIK3CA and PTEN at the top of the list”

Line 114: Please change “panel” to “range”

Line 120: The authors mention epithelial-to-mesenchymal transition increases invasiveness of a tumour. Based on single cell –omics it is not entirely clear that epithelial-to-mesenchymal transition actually occurs there is a hypothesis that it may represent a sampling error with residual stroma within the tumour when sampled. Its not necessary to go into detail in this review on this aspect but I would be cautious in how the phrase epithelial-to-mesenchymal transition is used and would suggested “possible epithelial-to-mesenchymal transition”

Line 130: One of several instances where a four digit number in this case 1929 is represented by a period instead of a comma, instead of 1.929 please use 1,929. There are other instances in the manuscript please review for same.

Line 135: “wild type patients” should be changed to “patients with wild type tumours”

Line 169: “apparition” should be changed to “the appearance of”

Line 177: “They” should be changed to “These therapies” to be clearer.

Line 192: Please rephrase

Line 221: “for futility” should be changed to “due to lack of efficacy” or at least to "as it was futile". I understand that the original authors used this terminology but I think the syntax is incorrect.

Line 231: Please change the word “signal”

Line 265: Please briefly explain “geometric Ki67” for readers not familiar with Ki67 as a biomarker. This is an important pathological predictor of behaviour.

Line 311: Please check spelling of transaminasitis versus transaminitis

Line 502: “higher rate of hyperamlasemia and hyperlipasemia”, could the authors confirm this is a reflection of an acute pancreatitis? The reference associated with this appears to be an abstract.

Line 570: One of a few instances where the word “adjunction” is used. Please use “in addition to” instead.

Line 607: “tumour biopsy” should be changed to “tumour biopsies”

Line 653: Again the word “adjunction” is used. Please use “addition” instead.

Line 672: “was” should be changed to “has been”

Author Response

Response to Reviewer 1 Comments

The authors present a review of PI3K inhibitors both pan-inhibitors and more specific sub-unit inhibitors. The review presents data concerning the fundamental associations between the PI3K molecular pathway and various cancers including breast cancer. The authors do an excellent job of then going on to discuss the state of the art in clinical deployment of PI3K inhibitors both in clinical practice and clinical trials. The review is timely, readable and comprehensive. Where appropriate, the authors also include conclusions and speculation around mechanisms of action, in particular around putative mechanisms of resistance, within the PI3K pathway.

We thank the Reviewer for his/her positive comments.

Although overall an excellent review and eminently readable, there are numerous minor errors in syntax which detract from the review and require amendment. In particular the words “the” and “a” or “an” are frequently left out where they should be used. The instances of this are too numerous to point out specifically and the review would benefit from correction in this area throughout the manuscript.

We have corrected these syntax errors.

The review could also benefit from the inclusion of a further figure indicating potential resistance mechanisms and how they could arise post-therapy in particular.

As suggested, we have added a figure 2 showing the potential resistance mechanisms and added references accordingly in the text in section “5. Resistance to PI3K inhibitors”

Other minor comments:

-Line 105: Please change “with at the top of the list PIK3CA and PTEN” to “with PIK3CA and PTEN at the top of the list”

We have modified the text.

-Line 114: Please change “panel” to “range”

We have modified the text.

Line 120: The authors mention epithelial-to-mesenchymal transition increases invasiveness of a tumour. Based on single cell –omics it is not entirely clear that epithelial-to-mesenchymal transition actually occurs there is a hypothesis that it may represent a sampling error with residual stroma within the tumour when sampled. It’s not necessary to go into detail in this review on this aspect but I would be cautious in how the phrase epithelial-to-mesenchymal transition is used and would suggested “possible epithelial-to-mesenchymal transition”

We thank the Reviewer for his/her comment and we have modified the text.

Line 130: One of several instances where a four digit number in this case 1929 is represented by a period instead of a comma, instead of 1.929 please use 1,929. There are other instances in the manuscript please review for same.

We have modified the text.

Line 135: “wild type patients” should be changed to “patients with wild type tumours”

We have modified the text.

Line 169: “apparition” should be changed to “the appearance of”

We have modified the text.

Line 177: “They” should be changed to “These therapies” to be clearer.

We have modified the text.

Line 192: Please rephrase

We have modified the text by “The main side effects observed during the study included nausea, hyperglycemia, rash, diarrhea, mucositis, depression and anxiety.”

Line 221: “for futility” should be changed to “due to lack of efficacy” or at least to "as it was futile". I understand that the original authors used this terminology but I think the syntax is incorrect.

We have modified the text.

Line 231: Please change the word “signal”

We have modified the text “an expected toxicity profile”.

Line 265: Please briefly explain “geometric Ki67” for readers not familiar with Ki67 as a biomarker. This is an important pathological predictor of behaviour.

We thank the Reviewer for his/her comment and we have modified the text with “Geometric Ki67 was used because of high variability in the determination of Ki67 and the approximate lognormal distribution of the data. Ki67 at day 15 were expressed as geometric mean proportions of the baseline and transformed into mean suppression (defined as one minus the geometric means of the proportional changes)”.

Line 311: Please check spelling of transaminasitis versus transaminitis

We have modified the text.

Line 502: “higher rate of hyperamylasemia and hyperlipasemia”, could the authors confirm this is a reflection of an acute pancreatitis? The reference associated with this appears to be an abstract.

We thank the Reviewer for his/her comment. Only data of the abstract were available. The authors used the CTCAE-v4 for grading the toxicity. According to this version, a grade 3 hyperamylasemia and hyperlipasemia corresponding to >2.0 - 5.0 x ULN unspecified concerning the symptomatology. In other studies with inavolisib, no cases of pancreatitis was describe.

Line 570: One of a few instances where the word “adjunction” is used. Please use “in addition to” instead.

We have modified the text as suggested.

Line 607: “tumour biopsy” should be changed to “tumour biopsies”

We have modified the text.

Line 653: Again the word “adjunction” is used. Please use “addition” instead.

We have replaced the word “adjunction” by “addition”.

Line 672: “was” should be changed to “has been”

We have modified the text.

Reviewer 2 Report

The manuscript by Bertucci et al provides a comprehensive overview of PI3K inhibitors for breast cancer treatment.

The manuscript is well written and reorts the results of the most recent and ongoing studies.

Minor comments:

- the authors could mention also the activation of RAS pathway in breast cancer, since it has a role in this cancer and is inserted in the figure.

-Hyperglicaemia is a relevant problem of PI3K inhibitors. The authors could emphasize this issue, since high glucose levels impair the whole metabolism, but also the crosstalk between breast cancer cells and surrounding cells (i.e. adipocytes and their precursors). For instance, Glucose enhances pro-tumorigenic functions  of Mammary Adipose-Derived Mesenchymal Stromal/Stem Cells and leads adipocyte production of cytokines, chemokines and growth factors (IL-8, CCL5, IGF1) driving breast cancer invasiveness and less responsive to tamoxifen.

--page 3, lines 91-101. Is this the figure 1 legend?

Author Response

Response to Reviewer 2 Comments

The manuscript by Bertucci et al provides a comprehensive overview of PI3K inhibitors for breast cancer treatment. The manuscript is well written and reports the results of the most recent and ongoing studies.

We thank the Reviewer for his/her positive comments.

Minor comments:

- the authors could mention also the activation of RAS pathway in breast cancer, since it has a role in this cancer and is inserted in the figure.

We thank the Reviewer for his/her comment and we have modified the text in section “5. Resistance to PI3K inhibitors”.

-Hyperglicaemia is a relevant problem of PI3K inhibitors. The authors could emphasize this issue, since high glucose levels impair the whole metabolism, but also the crosstalk between breast cancer cells and surrounding cells (i.e. adipocytes and their precursors). For instance, Glucose enhances pro-tumorigenic functions of Mammary Adipose-Derived Mesenchymal Stromal/Stem Cells and leads adipocyte production of cytokines, chemokines and growth factors (IL-8, CCL5, IGF1) driving breast cancer invasiveness and less responsive to tamoxifen.

We thank the Reviewer for his/her comment and we have modified the text “In addition to the impact on metabolism, an in vivo study showed that hyperglycemia could influence the BC tumor microenvironment. In co-culture in hyperglycemia com-pared to normal glycemia, adipocytes produced more inflammatory cytokines (IL-6, IL-8) and growth factors (VEGF, IGF1) and induced a more aggressive and invasive phenotype for BC cells in zebrafish

--page 3, lines 91-101. Is this the figure 1 legend?

Yes, the lines 91-101 represent the figure 1 legend, we followed the template of “Cancers”. But we have modified the text in order to avoid any confusion.

Reviewer 3 Report

This manuscript focuses on the efforts of developing pan or selective PI3K inhibitors for treatment of breast cancer. The manuscript is well written and provides a comprehensive view of the inhibitors currently under development or in clinical trials, and potential mechanism for resistance to these inhibitors. This manuscript will be of interest to the researchers in the field.

Author Response

Response to Reviewer 3 Comments

Reviewer 3's Comments and Authors’ Responses:

This manuscript focuses on the efforts of developing pan or selective PI3K inhibitors for treatment of breast cancer. The manuscript is well written and provides a comprehensive view of the inhibitors currently under development or in clinical trials, and potential mechanism for resistance to these inhibitors. This manuscript will be of interest to the researchers in the field.

Response: We thank the Reviewer for his/her positive comments.